# Approachability and Sensory Changes Following Mild Traumatic Brain Injury in Pigs

**DOI:** 10.3390/biomedicines12112427

**Published:** 2024-10-23

**Authors:** Mark Pavlichenko, Radina L. Lilova, Amanda Logan-Wesley, Karen M. Gorse, Audrey D. Lafrenaye

**Affiliations:** 1Department of Anatomy and Neurobiology, Virginia Commonwealth University, Richmond, VA 23298-0709, USA; 2Richmond Veterans Affairs Medical Center, Richmond, VA 23249-4915, USA

**Keywords:** traumatic brain injury, pig, fluid percussion injury, behavior, mechanical nociception, sensory, human approach task, approachability, von Frey

## Abstract

Background/Objectives: Traumatic brain injury (TBI) is a global healthcare concern affecting millions, with wide-ranging symptoms including sensory and behavioral changes that can persist long-term. Due to similarities with human brain cytoarchitecture and inflammation, minipig models are advantageous for translational TBI research. However, gaps in knowledge exist regarding their behavioral and sensory sequelae following injury. Methods: Therefore, in this study, we assessed changes in approachability using a forced human approach task (FHAT) and mechanical nociception using the von Frey test in adult male and female Yucatan minipigs for up to one week following a sham or central fluid percussion injury (cFPI). Specifically, the FHAT assessed each animal’s response to a forced interaction with either a known or unknown experimenter. To evaluate changes in nociceptive sensory sensitivity, von Frey monofilaments ranging from 0.008 to 300 g of force were applied to the pinna of the ear or base of the tail. Results: We found that forced approachability was affected by experimenter familiarity as well as cFPI in a sex-specific manner at subacute timepoints. We also found reductions in sensitivity following cFPI on the ear in male minipigs and on the tail in female minipigs. Conclusion: Overall, the current study demonstrates that cFPI produces both behavioral and sensory changes in minipigs up to one-week post-injury.

## 1. Introduction

Traumatic brain injury (TBI) is a leading health concern, with global estimates suggesting that 10 million people are annually hospitalized due to TBI [1]. In the United States, an annual average of approximately 53 thousand TBI-associated deaths occurred between 1997 and 2007 [2]. Both the long-term and short-term consequences of TBI are devastating to patients and their families. Significant negative consequences on physical health, mental health, and financial stability are common, with estimates of lifetime costs among US citizens reaching USD 40.6 billion in 2016 [3]. Although mild TBI (mTBI) accounts for at least 75% of all TBI, concerns exist that total estimates do not accurately capture the full scope of affected individuals [4].

The majority of patients with mTBI self-report the most severe symptoms occurring within the first 72 hours of injury, with improvements noticed after several days, and full recovery within weeks or months [5]. However, 5–20% of patients report persistent symptoms into the chronic phase following initial injury [6]. Behavioral changes, including those in sociability [7,8,9] and sensory sensitivity [10,11,12], are some of the many consequences that can affect patients following TBI.

Translational animal models are essential tools for investigating changes in behavior after injury, affording a means to investigate the causal relationship between TBI and functional deficits [13]. The pig model has emerged as a leading translational model of TBI, largely due to pig brains’ similarity to human brains in terms of cytoarchitecture, anatomy, metabolic rates, and inflammatory responses [14]. However, there remains a dearth of knowledge regarding behavioral sequelae following TBI in the pig, which spans various injury severities, assessed modalities, and behavioral tests [13]. Particularly apparent are the lack of investigations into sociability/approachability and sensory changes following mTBI, a condition in which diffuse brain pathologies prevail [15].

Therefore, this study assessed approachability and mechanical nociception in adult male and female minipigs up to one week following a diffuse mTBI. Approachability, a component of sociability, was assessed using an in-pen, forced human approach task (FHAT), in which the minipig’s response to a forced interaction with either a known or unknown human was evaluated. Changes in mechanical nociception were evaluated using von Frey monofilaments of various strengths, which are commonly used in both experimental and clinical assessments of tactile sensitivity [16,17,18,19,20]; however, von Frey filaments have not previously been used in pigs following TBI. All assessments were performed on freely moving minipigs in their home runs. We hypothesized a reduction in approachability following mTBI, with a more severe aversion to forced interaction with an unknown versus known investigator. Based on consistent findings of increased sensitivity to von Frey mechanical stimulation in rodent studies [16,18,20,21,22], we further hypothesized that minipigs would exhibit hypersensitivity to mechanical nociception sub-acutely following mTBI. Ultimately, this study investigated two novel methods of behavioral assessment post-TBI that can be achieved in-pen without specialized equipment. We found both sensory and approachability changes at one week post-injury, which indicates that these behavioral modalities are sensitive to subtle changes induced by mild TBI in a translational pig model.

## 2. Materials and Methods

### 2.1. Animals

A total of 12 castrated male and 14 normally cycling female adult minipigs (~6 months old; 21.5–43 kg) from either Sinclair Bio Resources (Auxvasse, MO, USA) or Premier BioSource (Ramona, CA, USA) were used for these studies. Two animals (one male and one female) were euthanized prior to the scientific endpoint of the study due to TBI-related recovery issues. All studies were approved by and conducted under the supervision of the Richmond Veterans Affairs Medical Center’s Institutional Animal Care and Use Committee in accordance with guidelines consistent with The Guide for the Care and Use of Laboratory Animals as well as United States Department of Agriculture (USDA) regulations Center (Protocol number: 1715187 Approval Date: 19 January 2023) [23,24,25]. Minipigs had shared housing in one large pen unit prior to sham or injury induction and separate housing in individual neighboring runs thereafter. Animals were kept on a 12 h light–dark cycle, had free access to water, and received two pounds of normal diet (Teklad miniswine diet 8753, Inotiv, Madison, WI, USA) daily. Minipigs were housed in mixed sham/injured same-sex cohorts of three animals per group. Pen units consisted of five identical 60” × 36” × 66” runs (Allentown Inc., Allentown, NJ, USA), each with a front entrance and with the ability to be interconnected or separated, as needed.

Videos were recorded for all behavioral tests conducted in this study using Noldus MediaRecorder software (Noldus, Leesburg, VA, USA). All animals were randomly assigned to the sham or TBI groups prior to the initiation of the study. All behavioral assessments were done prior to sham or injury then at 1 d, 2 d, 3 d, and 1 w post-injury in the following order: (1) forced human approach task (FHA; unknown experimenter first, known experimenter second) followed by (2) the von Frey test (Figure 1).

### 2.2. Surgical Preparation and Induction of Traumatic Brain Injury

Surgical preparation and injury induction were done as previously described [26]. Briefly, initial anesthesia was induced via an intramuscular injection of ketamine (2–5 mg/kg dose; Covetrus, Portland, ME, USA) and tiletamine/zolazepam (2 mg/kg dose; Zoetis, Parsippany, NJ, USA). To decrease salivation prior to intubation, glycopyrrolate (0.025 mg/kg; Exela Pharma Sciences, Lenoir, NC, USA) was administered intramuscularly. Propofol (3 mg/kg; Sagent Pharmaceuticals, Schaumberg, IL, USA) was administered prior to intubation with a 6–7 mm endotracheal tube (Well Lead Medical Co., Guangzhou, China). Anesthesia was maintained with 1–2.5% isoflurane (Baxter, Deerfield, IL, USA) in a 70:30 room-air-to-oxygen ratio for the duration of the procedure. An intravenous (IV) 18–22-gauge cannula (Becton, Dickinson, and Company, Franklin Lakes, NJ, USA) was placed in the ear for fluid replacement with Lactated Ringer’s Solution at a rate of 5 mg/kg/hr for the duration of the procedure. A BMDSTM transponder temperature microchip (TP-1000; Avidity Science, Waterford, WI, USA) was inserted into subcutaneous adipose tissue at the base of the neck. Physiological monitoring of temperature, blood pressure, blood oxygen saturation, heart rate, end-tidal carbon dioxide, respiratory rate, and electrocardiogram was conducted and the results were recorded every 15 min for the duration of the procedure.

Following sterile surgical site isolation, an initial incision was made from the supraorbital process to the nuchal crest and the surface of the skull was exposed. A 16 mm Galt skull trephine was used to create a craniectomy centered 21 mm anterior to the nuchal crest along the sagittal suture while leaving the dura mater intact. A custom stainless-steel threaded hub (Custom Design and Fabrication, Richmond, VA, USA) with an outer diameter of 17 mm and an inner diameter of 14 mm was screwed into the craniectomy site to a depth of ~4 mm at an angle perpendicular to the surface of the skull. The animal was then carefully positioned so that the hub could be affixed to an L-shaped adapter on the fluid percussion device in preparation for central fluid percussion injury (cFPI) induction. At the time of injury, a 4.8 kg steel pendulum was released onto the fluid percussion device to induce a fluid pulse of 1.82 ± 0.3 atm (~0.09 atm/kg) for 25–35 ms through the intact dura. The magnitude and duration of the injury were recorded via an oscilloscope (Tektronix, Beaverton, OR, USA). Sham minipigs received identical preparations up to and including connection to the fluid percussion device, but the pendulum was not released. Following sham or cFPI, the hub was removed, the scalp was fully sutured with polyglactin sutures (Ethicon, Raritan, NJ, USA), and a mixture of antibiotic (Perrigo, Allegan, MN, USA) and lidocaine (IMS Limited, South El Monte, CA, USA) was applied to the incision site. Extended-release buprenorphine-HCl (Wedgewood, Swedesboro, NJ, USA) was administered subcutaneously 15 min post-injury and prior to recovery for post-operative pain control.

At 1 w following injury, animals were anesthetized then were euthanized via an intravenous dose of pentobarbital sodium and phenytoin sodium solution (1 mL/4.5 kg; Vortech, Dearborn, MI, USA), with death confirmed via cardiac auscultation.

### 2.3. Forced Human Approach Task (FHAT)

To gauge the approachability of the minipigs to humans, a forced human approach task (FHAT) was performed, in which the response to a forced interaction with a “known” or “unknown” human was assessed. To establish the known/unknown human paradigm for the FHAT, following the initial 1 w quarantine period, the minipigs became acquainted with one experimenter. This acquaintance period consisted of one consistent experimenter visiting the minipigs for upwards of 1 h per day for 1 w prior to injury induction (Figure 1). Initial acclimation consisted of the known experimenter remaining outside of the run, petting the minipigs’ snout and giving the pig snacks different from their everyday diet. Once the minipigs became familiar and more comfortable with the known human, he/she would enter the run, pet the animals and rub them with mineral oil to get them used to physical interaction, and give them snacks. The “unknown” experimenter was another experimenter who did not participate in this period of familiarization with the animals. The same known and unknown experimenters were used for all days of testing.

For the FHAT assessment, the experimenter entered the pen and slowly approached the pig with an outstretched hand. The experimenter would then touch the animal on three distinct loci: neck, back, and rump, in that order. Animals’ reactions to touch at all three loci as well as overall demeanor were verbalized by the experimenter. Experimenters remained in the runs long enough to sufficiently perform the FHAT before exiting the run. The unknown experimenter performed the FHAT first followed by a known experimenter for all testing sessions (Figure 1).

Recorded FHAT sessions were scored on a 6-point scale assessing the minipig’s reactions to physical touch, with 1 being the least approachable and 6 being the most approachable (Table 1). A score of 0 was reserved for any minipig unable to ambulate either away or towards the experimenter with only one pig receiving this score. All videos were blinded for timepoint and injury prior to FHAT analysis.

### 2.4. Von Frey Monofilament Tests

Von Frey monofilament tests were carried out to assess changes in somatosensation in minipigs following sham or cFPI-induced mTBI (Figure 1). A set of 20 retractable plastic monofilaments based on the Semmes Weinstein set, ranging from 0.008 to 300 g of force (Bioseb, Pinellas Park, FL, USA), was used to mechanically stimulate specific loci on the minipigs. The pinna of the ear that did not have the USDA tag on it and the base of the tail were the two assessed loci in the current study. While not overtly assessing experimenter-minipig familiarity effects, to maintain consistency for the FHAT assay, the known experimenter performed the von Frey test while the unknown experimenter helped keep the minipigs’ attention in one location by feeding them treats from outside the run. Responses were assessed by the experimenter inside the run and verbalized for note taking by the unknown experimenter as well as for the video recording concurrently running.

Briefly, each locus of stimulation was assessed until a withdrawal threshold was reached. Experimenters started with a 60 g monofilament on the ear and, subsequently, on the tail, increasing or decreasing monofilament strength depending on presence or lack of response. A response for the ear was defined as a definitive flick in response to the monofilament. A response for the tail was defined as definitive flicking or tucking in response to the monofilament. The lowest strength monofilament to elicit a response on each locus twice was recorded as the minipig’s response threshold for that locus. If the maximum strength filament (300 g) failed to elicit a response, the minipig was noted as “non-responsive” (NR) to mechanical stimulation at that particular locus.

### 2.5. Statistical Analysis

Analysis was conducted in R (R4.3.3; R Core Team, 2024). The Shapiro–Wilkes test was used to assess for data normality. Neither data set was normally distributed; therefore, non-parametric paired Wilcoxon comparisons, Mann–Whitney U, and Kruskal–Wallis tests with a Dunn’s post hoc test were used. A significance threshold of *p* < 0.05 was set for all analyses. Error bars are depicted as standard error of the mean (S.E.M.).

## 3. Results

### 3.1. Forced Human Approach by an Unknown Experimenter Is Negatively Affected by cFPI in Female Minipigs

The FHAT was used to investigate changes in a minipig’s approachability to a human experimenter after either sham or TBI induced via cFPI. In male minipigs, FHAT scores were not significantly affected by cFPI or experimenter familiarity at any timepoint post-injury (Figure 2A,B and Appendix A). Approachability to the known experimenter was consistent between sham and TBI female at all post-injury timepoints (Figure 2C). However, in female minipigs receiving a cFPI, approachability was decreased at the 1 w timepoint compared to sham females when approached by an unknown experimenter (z = −2.03, W = 34.5, *p* = 0.0418; Figure 2D and Appendix A).

### 3.2. Von Frey Sensitivities Are Affected by TBI in Sex-Dependent Loci

The von Frey monofilament test was used to assess changes in mechanical nociception on the ear and base of the tail following injury. Sex- and location-dependent decreases in mechanical sensitivity were observed post-TBI. Specifically, male TBI minipigs were less sensitive to von Frey stimulation of the ear post-cFPI when compared to pre-injury baseline (*X*^2^(4) = 14.5, *p* = 0.006; Figure 3A). This reduced responsiveness to von Frey stimulation compared to pre-injury baseline was found at 1 d (*p* = 0.0019), 2 d (*p* = 0.015), 3 d (*p* = 0.0019), and 1 w (*p* = 0.0019) post-cFPI. Further, TBI males exhibited lower sensitivity in the ear at 3 d compared to timepoint-matched sham controls (z = −1.97, W = 6, *p* = 0.048; Figure 3A and Appendix A). There was a significant interaction between injury and timepoint on von Frey responses in the male ear (*X*^2^(9) = 19.7, *p* = 0.02). The responsiveness to von Frey monofilament stimulation in the tail of male pigs, however, was consistent across the pre- and post-sham timepoints (Figure 3B). Meanwhile, female TBI minipigs displayed decreased tail sensitivity compared to pre-injury responses (*X*^2^(4) = 13.1, *p* = 0.011) at 1 d (*p* = 0.014), 2 d (*p* = 0.032), 3 d (*p* = 0.027), and 1 w (*p* = 0.0004) post-injury (Figure 3D). TBI females exhibited a significant decrease in tail sensitivity at 1 w compared with timepoint-matched sham animals (z = −2.38, W = 7.0, *p* = 0.018; Figure 3D and Appendix A). The responsiveness to von Frey monofilament stimulation in the ear of female pigs, however, was consistent across the pre- and post-sham timepoints (Figure 3C).

## 4. Discussion

This study aimed to investigate changes in social behaviors and somatosensation over a 1 w, subacute timeframe following cFPI-induced TBI in male and female minipigs. A minipig model was selected for its higher degree of similarity to human brain anatomy [27,28], cytoarchitecture [27], ratios of grey matter to white matter [28,29,30], and neuroinflammation [31], which promotes increased translatability [14]. Approachability, a component of sociability, was assessed via modified in-pen FHAT, and changes in mechanical somatosensation were assessed via the von Frey monofilament test.

The FHAT was used as an invasive measure of approachability in which human investigators entered the minipigs’ space and initiated contact with the animals. This interaction could create an avoidance reaction, thus providing insight into an animal’s overall emotional valence [32]. A similar test has been used by farmers to assess animal welfare and the human–animal relationship in which minipigs were assessed for their degree of withdrawal from the experimenter, characterized by distance retreated and loci the minipig could be touched on [32,33,34]. Our FHAT in these studies was similar to the farm paradigm but modified depending on our physical restrictions. While each run individually housed a single minipig for testing, the neighboring runs were separated by bars and were all in the same room, which allowed minipigs to see and hear other animals. Additionally, due to single run size constraints, we could not feasibly assess the distance withdrawn. Rather, minipigs were assessed based on the experimenter’s ability to make physical contact with them and the minipig’s behavior in response to this contact on three distinct loci: neck, back, and rump.

Female TBI minipigs demonstrated a significant reduction in approachability when interacting with an unknown experimenter at 1 w post-cFPI relative to female sham animals interacting with an unknown experimenter at the same 1 w timepoint (Figure 2B). This response was not seen in female TBI minipigs interacting with a known experimenter (Figure 2A) or in male minipigs regardless of experimenter familiarity (Figure 2). It was hypothesized that FHAT scores would significantly decrease during interactions with an unknown compared to known experimenter, regardless of sex or injury. While our findings did not fully support this hypothesis, the data do provide indications that TBI and experimenter familiarity can concomitantly result in a sex-specific decrease in approachability at more subacute timepoints after injury.

In clinical studies, mTBI has been associated with discernible negative changes in mental health [35]. A study published in the American Journal of Psychiatry surveyed patients with mTBI and found that injury was associated with elevated odds of developing a panic disorder, social phobia, and/or agoraphobia [7]. Additional studies have demonstrated a higher prevalence of affective disorders such as major depressive disorder in mTBI patients compared to the general population [36]. Dysfunctions of social behavior have been noted to be prevalent in TBI patients, with studies reporting that patients encountered problems with social behaviors across all TBI severities [8,9]. In the current study, we found reduced approachability following TBI, but only in female cohorts at 1 w post-injury. A potential explanation for this observation is that, over the 1 w testing period following injury, minipigs may have become more acclimated to the unknown experimenter. And while all sham minipigs and male TBI minipigs may have become acclimated to the unknown experimenter, our findings would suggest an impairment in this acclimation in female TBI minipigs. Regardless, this sex-specific difference in approachability warrants further investigation to validate and potentially expand on.

In patients suffering with TBI, aggression is a prevalent behavior, with one study classifying up to 25% of assessed patients as aggressive [8] while another identified verbal aggression and inappropriate social behavior as the most prominent maladaptive behavior in patients with brain injury [37]. In our study, while some minipigs did demonstrate aggressive behaviors during FHAT, the majority of animals demonstrated more evasive behaviors. This lack of aggression could be attributed to the animal strain itself, as Yucatan minipigs are bred for smaller size and affability. Additionally, male animals were castrated, also potentially influencing aggressive tendencies. Therefore, while hormonal fluctuations might play a role in TBI-induced aggression and other social changes after injury, further studies would be needed to interrogate this possibility.

In addition, external stressors could have influenced approachability independent of injury, sex, or experimenter familiarity. While each cohort of three animals was uniform in sex, cohorts consisted of a mixture of injured and sham animals to mitigate potential environmental confounders between cohorts. However, the possibility that environmental stressors varied across cohorts cannot be excluded. Further, as approachability changes were only seen at the latest timepoint of this study (1 w); future investigations into behavioral changes occurring at more chronic timepoints are warranted.

While the number of animals used for each group in this study was properly powered to investigate differences between TBI and sham, we noted a higher degree of variability between animals than originally anticipated (Appendix A). This could have had an outsized impact on statistical trends that would potentially be dampened in studies with larger numbers of animals. These data, however, are still valuable, as this is the first study to investigate FHAT and von Frey responses within the first week in minipigs following a mild TBI and will facilitate more specific a priori analyses in the future.

We also performed von Frey monofilament sensitivity assessments on the pinna of the ear and on the base of the tail in awake, freely moving minipigs. The home-pen environment was selected to reduce anticipated stress from moving the minipigs to a separate testing environment, potentially impacting approachability assessments. The ear and tail were chosen due to their independent mobility, physical accessibility to researchers, and functional relevance to pig behavior [38,39]. The ear has been shown to be a locus of nociceptive response in prior studies, with a higher occurrence of ear flicking or flapping in piglets following ear notching [40]. Ear flicking has also been identified as a positive response during assessments of tactile allodynia in the minipig model of neurofibromatosis type 1 [41]. Tail posture and movement are also salient indicators for a pig’s emotional valence and state of arousal, namely, based on tail curling, tucking, and/or wagging [38]. Variability in ear and tail movements was beneficial in discriminating responses specific to the monofilament rather than baseline tail-wagging or ear movement or response to an external stimulus, e.g., a loud noise.

Because minipigs were freely moving in their home pens during behavioral assessments, the von Frey test required two experimenters: one to perform monofilament testing and another to feed the minipig treats to keep its head and body in a relatively consistent location. While this approach afforded some consistency in positioning for the duration of testing, eating the treats would occasionally result in some rapid movements of the head and tail wagging. To avoid this potential confound, experimenters waited until the ear or tail was not moving before applying the mechanical stimulation. All stimulations were done in triplicate, and a response was recorded when the animal responded at least two out of three times to that monofilament strength. While a definitive ear flick was defined as a response to mechanical stimulus, experimenters noted that some minipigs would ‘brace’ their ears backwards and closer to their heads after the filament was tested rather than flick them. Such a response was documented but not counted as a response to stimulation. Backward-pointed ear positioning has been shown to occur at higher rates in response to aversive stimulation, indicating a negative valence [39], thus indicating a potentially different but aversive response to von Frey stimulation that would require more investigation in future studies.

Male and female TBI minipigs had significantly reduced sensitivity scores at all assessed post-injury timepoints compared to pre-injury scores in a location- and sex-specific manner. Male minipigs displayed hyposensitivity in the ear while female minipigs displayed hyposensitivity in the tail following TBI (Figure 3). While this study was not powered to specifically investigate sex differences, the current data indicate a potential influence of sex in TBI-induced sensitivity changes, thus warranting further studies powered to resolve such potential effects.

The observed reduction in minipig ear and tail sensitivity following TBI is contrary to findings in rodent models of TBI, in which sensitivity in multiple loci increases following injury [18,20]. Somatosensory hypersensitivity observed in rodents post-TBI has been associated with increased inflammation in the sensory cortex [28,30,42,43]. In clinical studies, the consequences of TBI include varied atypical sensory sensitivities [10]. Changes in sensitivity span a wide range of modalities such as light [44,45], auditory [46,47], olfactory [48], gustatory [12], and vestibular perception [49], with a majority of studies identifying hypersensitivity following injury [10]. Brain injury has also been linked to changes in mechanical nociception, with reports of post-injury hypersensitivity and tactile anomalies [11,12]. However, there is still a gap in the literature regarding changes in tactile sensitivity following TBI, especially in mTBI with primarily diffuse pathologies.

Due to the high degree of variability between pigs (Appendix A), it is difficult to draw firm conclusions regarding the observed decreased von Frey sensitivities in the current study. Interestingly, a study that utilized the von Frey test to examine the impact of prenatal stress on nociception in juvenile pigs found that stressed groups exhibited less sensitivity to mechanical stimulus compared to non-stressed controls [50]. Coupled with our findings, this suggests hyposensitivity to mechanical nociception could be a common and expected response in minipigs exposed to injury and/or stress. Due to the lack of publications using von Frey sensitivity assessments in minipigs, however, the findings remain difficult to validate.

It is, further, possible that the observed reduction in von Frey monofilament responses might not signify reductions in sensitivity. It is possible that during stimulation of the ear, the monofilaments could have contacted hairs growing in the ear, potentially causing the response rather than the monofilament contact with the pinna itself. If this were the case, however, we would have expected observed hypersensitivity rather than hyposensitivity. Additionally, the lack of response to high-gauge von Frey filaments in both the ear and tail of TBI pigs could indicate that the pig’s attention was more easily diverted away from the sensory stimulus and toward the experimenter feeding them snacks. A lack of response to von Frey monofilament stimulation could potentially also indicate an indifference to a stimulus that is perceived as non-dangerous; however, the fact that this was not seen in sham animals reduces the likelihood that this is a normal response. Future studies could be designed to eliminate potential confounds of attention diversion by restraining the pig during von Frey testing. Future studies could also investigate additional stimulus loci, such as the periorbital region or the tip of the snout, to determine if von Frey filament responsiveness might differ across other regions in a sex-dependent manner as well.

## 5. Conclusions

Overall, we found changes in social behavior and somatosensation after cFPI-induced diffuse mTBI. Approachability in a forced animal–human interaction paradigm with an unknown experimenter was lower in TBI female minipigs compared to sham females, a result not recapitulated in male minipigs. Mechanical sensitivity assessed via von Frey monofilament tests found reduced sensitivity in minipigs after TBI in different loci that differed by sex. Male minipigs demonstrated hyposensitivity on the ear while females demonstrated hyposensitivity on the tail.

This study aimed to fill gaps in knowledge of changes in certain social behaviors and sensory perceptions in a higher-order animal model of diffuse mTBI. A stronger understanding of diffuse-TBI-associated behavioral and sensory sequelae in higher-order animal models is needed, with few validated and published behavioral studies relative to more established rodent models.

Ultimately, this study marks the first application of the FHAT and von Frey somatosensory sensitivity test in a minipig model of diffuse mild TBI. While the physiological translatability of pigs to human brain pathology has become more established, work is still needed to develop and validate specific behavioral assays in pig models of TBI. Further, future studies should be done to associate these behavioral changes with quantitative pathological and molecular data in order to elucidate the underlying mechanisms that evoke altered behavioral and sensory outcomes following traumatic brain injury. As there were significant changes observed at 1 w post-injury, investigations into the progression of these behavioral changes over longer post-injury timepoints would also be warranted. Finally, as there are so few investigations into behavioral changes following TBI in the pig, the findings presented in the current study should be replicated and expanded upon in future studies.

## Figures and Tables

**Figure 1 biomedicines-12-02427-f001:**
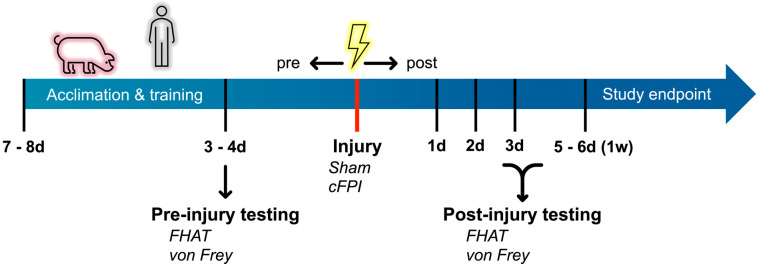
Experimental timeline for the current study. Acclimation to a known experimenter was done for a week following animal release from quarantine. A pre-injury behavioral assessment was done ~3d prior to induction of a central fluid percussion injury (cFPI) or a control sham injury (red line). Post-injury behavioral assessments were performed at 1 d, 2 d, 3 d, and 1 w (5–6 d) post-injury. During all behavioral testing sessions, the FHAT was performed prior to the von Frey test. Following completion of all behavioral assessments, animals were euthanized.

**Figure 2 biomedicines-12-02427-f002:**
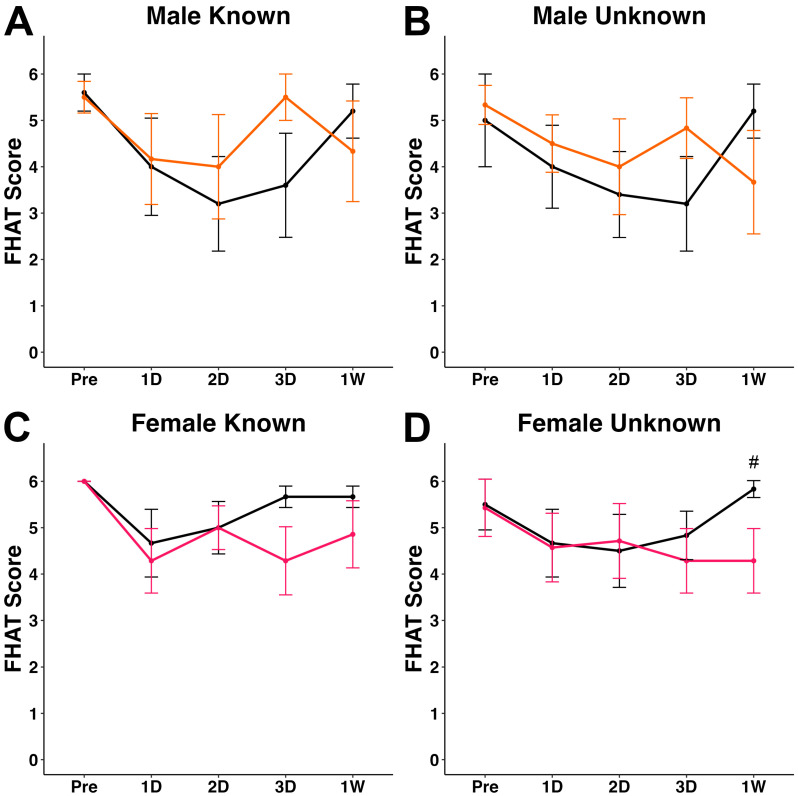
Plots of mean forced human approach test (FHAT) score for (**A**,**B**) male sham (n = 5, grey lines) and TBI (n = 6, orange lines) and (**C**,**D**) female sham (n = 6, grey lines) and TBI (n = 7, pink lines. Each FHAT test was performed by both a (**A**,**C**) known and (**B**,**D**) unknown experimenter. # *p* < 0.05 compared to timepoint-matched sham.

**Figure 3 biomedicines-12-02427-f003:**
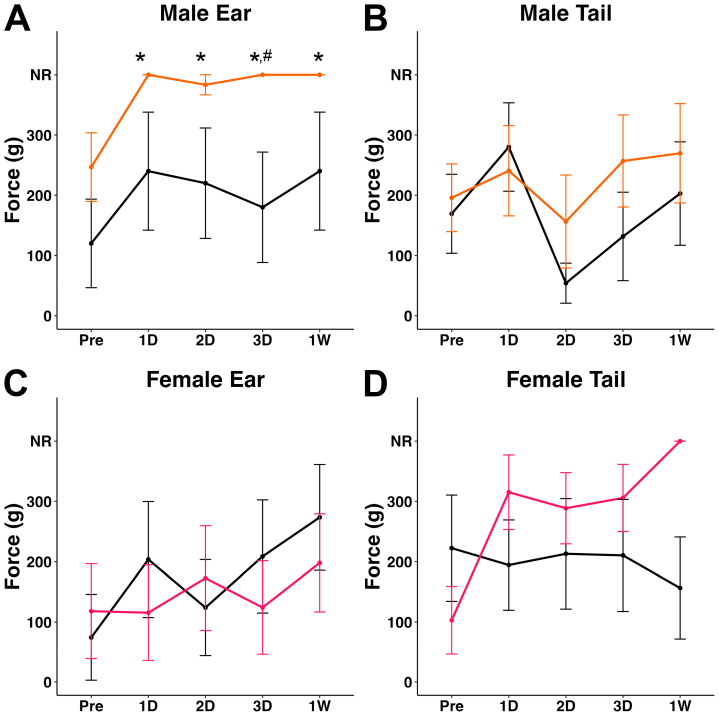
Plots of mean von Frey filament gram force that elicited a response in the (**A**,**C**) ear or (**B**,**D**) tail for (**A**,**B**) male or (**C**,**D**) female pigs prior to a TBI and up to 1 w following TBI. Male sham (n = 5) and female sham (n = 5) pigs are indicated as black lines. Male (n = 6) and female (n = 7) pigs sustaining a TBI are indicated as orange lines for males and pink lines for females. NR = no response to the highest gram force filament (300g). * *p* < 0.05 compared to pre-injury, # *p* < 0.05 compared to time-point matched sham.

**Table 1 biomedicines-12-02427-t001:** Scoring rubric for the forced human approach task (FHAT). Minipig behavioral descriptions corresponding to FHAT scoring from 1 (least approachable) to 6 (most approachable). A score of 0 was reserved for animals unable to ambulate either away from or towards the experimenter.

FHAT Score	Minipig Behavioral Description
0	Unable to interact with or move away from experimenter
1	Avoidance of any contact and/or display of defensive behaviors (e.g., growling)
2	Withdrawal upon any physical contact
3	Withdrawal upon physical contact on more than one locus
4	Tensing/shying upon physical contact on certain loci; no withdrawal
5	No withdrawal or tensing upon physical contact; no interaction with experimenter
6	No withdrawal or tensing upon physical contact; interaction with experimenter (e.g., voluntarily approaching experimenter, rubbing body against experimenter’s legs, chewing shoelaces/shoe covers)

## Data Availability

Full data for each animal can be found in Appendix A.

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
