# Peer review of "Approachability and Sensory Changes Following Mild Traumatic Brain Injury in Pigs"

_biomedicines, 2024, doi:10.3390/biomedicines12112427_

Round 1
Reviewer 1 Report
Comments and Suggestions for Authors
After the added changes, I partially agree with the authors remarks.
Author Response
After the added changes, I partially agree with the authors remarks.
OK
Reviewer 2 Report
Comments and Suggestions for Authors
The paper is well written and within the scope of the journal. It is of the interest for the readers and generally fulfills the quality norms for the journal paper. Only minor improvements can be suggested to the authors, but in my view the paper should be published. The suggested improvements are the following:
* Although mostly clear the title of the paper appears too long and the authors are suggested to try to shorten it if possible without loosing the essence, which contributes to the increased citation of the paper in the future.
* Like title, keywords should be reconsidered for shortening - for example mild traumatic brain injury, or just brain injury, etc. Shorter and more clear and wider keywords makes paper citations easier in the future.
* Last paragraph of the Introduction is devoted to the contents of the paper and generally valuable. Authors are encouraged to express novelty and contribution more clearly and directly, by expanding this last paragraph in introduction or creating another short following paragraph.
* It might be recommended to add comment on if and how the limited number of subjects in the study, which is understandable, affects presented results.
* Please try to avoid mutiple refencing,, for example [27-32] line 244, which makes more difficult to the readers to isolate claims.
* Figure 4 is referenced in the text but missing.
* In Conclusions section please expand future directions to a separate short subparagraph.
* Table 1 appears to have two captions, both above and below Table.
* Please try to improve graphical quality of the figures, especialy of the figure 1.
Comments on the Quality of English Language
The English is generally OK and the paper is clear only minor improvements and final proofreading could be suggested if the paper reaches publication stage.
Author Response
* Although mostly clear the title of the paper appears too long and the authors are suggested to try to shorten it if possible without loosing the essence, which contributes to the increased citation of the paper in the future.
We have now shortened the title for clarity and brevity. “Approachability and Sensory Changes Following Mild Traumatic Brain Injury in Pigs ”
* Like title, keywords should be reconsidered for shortening - for example mild traumatic brain injury, or just brain injury, etc. Shorter and more clear and wider keywords makes paper citations easier in the future.
We have adjusted some of our key words to make it easier for people to find the paper. These keywords are now: “traumatic brain injury; swine; pig; fluid percussion injury; behavior; mechanical nociception; sensory; human approach task; approachability; von Frey”
* Last paragraph of the Introduction is devoted to the contents of the paper and generally valuable. Authors are encouraged to express novelty and contribution more clearly and directly, by expanding this last paragraph in introduction or creating another short following paragraph.
We have now expanded the final paragraph of the introduction to incorporate a clearer and direct commentary on the novelty and efficacy of these behavioral tests in swine. “Ultimately, this study investigated two novel methods of behavioral assessment post-TBI that can be achieved in-pen without specialized equipment. We found both sensory and approachability changes at 1-week post-injury, which indicates that these behavioral modalities are sensitive to subtle changes induced by mild TBI in a translational swine model. ”
* It might be recommended to add comment on if and how the limited number of subjects in the study, which is understandable, affects presented results.
We have now included a paragraph discussing the low number of animals. “While the number of animals used for each group in this study was properly powered to investigate differences between TBI and sham, we noted a higher degree of variability between animals than originally anticipated (supplemental figure S1). This could have had an outsized impact on statistical trends that would potentially be dampened in studies with larger numbers of animals. This data, however, is still valuable, as it is the first study to investigate FHAT and Von Frey responses within the first week in minipig following a mild TBI and will facilitate more specific A Priori analyses in the future. ”
* Please try to avoid multiple refencing, for example [27-32] line 244, which makes more difficult to the readers to isolate claims.
We have now limited the number of multiple citations where possible.
* Figure 4 is referenced in the text but missing.
This was a typo and should have referenced figure 3. We have fixed this.
* In Conclusions section please expand future directions to a separate short subparagraph.
We have now expanded the final paragraph in the conclusion “Ultimately, this study marks the first application of the FHAT and von Frey somatosensory sensitivity test in a minipig model of diffuse mild TBI. While the physiological translatability of pigs to human brain pathology has become more established, work is still needed to develop and validate specific behavioral assays in pig models of TBI. Further, future studies should be done to associate these behavioral changes with quantitative pathological and molecular data in order to elucidate the underlying mechanisms that evoke altered behavioral and sensory outcomes following traumatic brain injury. As there were significant changes observed at 1w post-injury, investigations into the progression of these behavioral changes over longer post-injury timepoints would also be warranted. Finally, as there are so few investigations into behavioral changes following TBI in the pig, the findings presented in the current study should be replicated and expanded upon in future studies.”
* Table 1 appears to have two captions, both above and below Table.
We have removed the top table title to avoid confusion.
* Please try to improve graphical quality of the figures, especialy of the figure 1.
We have removed the colors on the significance indicators on Figures 2 and 3. We have also completely redone Figure 1.
Reviewer 3 Report
Comments and Suggestions for Authors
Key Points
- Impact of Traumatic Brain Injury (TBI): Traumatic brain injury can lead to long-lasting sensory and behavioral changes.
- Advantages of Minipig Models: Minipigs are beneficial for TBI research due to their similarities to human brain structure and inflammatory responses.
- Knowledge Gaps: There is insufficient understanding of the behavioral and sensory changes in minipigs following injury.
- Research Objective: This study aims to evaluate changes in approachability and mechanical nociception in minipigs after either a sham procedure or central fluid percussion injury (cFPI).
Useful Aspects
- Forced Human Approach Task (FHAT): This task assesses how each minipig reacts to forced interactions with either a familiar or unfamiliar experimenter.
- von Frey Test: This test measures changes in nociceptive sensory sensitivity by applying varying forces (from 0.008 to 300 grams) to the ear or the base of the tail.
- Impact of Sex Differences: The study found that both experimenter familiarity and cFPI influence approachability in a manner that varies by sex.
- Sensory Changes: The research revealed a decrease in sensitivity following cFPI, with male minipigs showing reduced sensitivity in the ear and female minipigs in the tail.
Comparison with Past Academic Cases
- Comparison with Human and Animal Models: Studies on human TBI have reported sensory and behavioral changes that align with the findings observed in minipig models.
- Impact of Sex Differences: Previous research has also indicated that the effects of TBI differ by sex, which is consistent with the results of this study.
- Evaluation of Nociception: The von Frey test is a widely accepted method in other animal models for assessing sensory changes, establishing it as a standard approach.
This study offers valuable insights into TBI research utilizing minipig models and may aid in the development of future treatment strategies.
Additional Points and Issues to Discuss
Evaluation of Long-term Effects:
This study evaluates changes up to one week post-TBI, but there is a lack of data on long-term effects. Further research is needed to assess long-term behavioral and sensory changes.
Introduction of Other Behavioral Tests:
In addition to FHAT and von Frey tests, incorporating other behavioral tests (e.g., open field test, maze test) could provide a more comprehensive behavioral assessment.
Assessment of Pathological Correlations:
It is important to evaluate the correlation between behavioral and sensory changes and brain pathological changes. This can provide a deeper understanding of the mechanisms of TBI.
The following paper may be helpful, although it is not an mTBI model, but a radiation model.
Yoneoka, Y., Satoh, M., Akiyama, K., Sano, K., Fujii, Y., & Tanaka, R. (1999). An experimental study of radiation-induced cognitive dysfunction in an adult rat model. The British journal of radiology, 72(864), 1196–1201. https://doi.org/10.1259/bjr.72.864.10703477
Comments on the Quality of English Language
Please review for minor spelling mistakes and ensure terminology consistency.
Author Response
Introduction of Other Behavioral Tests:
In addition to FHAT and von Frey tests, incorporating other behavioral tests (e.g., open field test, maze test) could provide a more comprehensive behavioral assessment.
We agree with the reviewer that additional behavioral tests in pigs following TBI are needed. This study, however, focused on two behavioral tests.
Assessment of Pathological Correlations:
It is important to evaluate the correlation between behavioral and sensory changes and brain pathological changes. This can provide a deeper understanding of the mechanisms of TBI.
We agree with this statement. We have included such a statement in our manuscript: “While the physiological translatability of pigs to human brain pathology has become more established, work is still needed to develop and validate specific behavioral assays in pig models of TBI.”
The following paper may be helpful, although it is not an mTBI model, but a radiation model.
Yoneoka, Y., Satoh, M., Akiyama, K., Sano, K., Fujii, Y., & Tanaka, R. (1999). An experimental study of radiation-induced cognitive dysfunction in an adult rat model. The British journal of radiology, 72(864), 1196–1201. https://doi.org/10.1259/bjr.72.864.10703477
Thank you for the suggestion of study citation. After reading this article, however, this work does not fit with the current study in timing, assessments done, or species. We therefore, feel this citation would not fit for the current manuscript.
Comments on the Quality of English Language
Please review for minor spelling mistakes and ensure terminology consistency.
We have verified that spelling and terminology are consistent throughout the manuscript.